# Immune Response to Biofilm Growing Pulmonary *Pseudomonas aeruginosa* Infection

**DOI:** 10.3390/biomedicines10092064

**Published:** 2022-08-24

**Authors:** Kim Thomsen, Niels Høiby, Peter Østrup Jensen, Oana Ciofu, Claus Moser

**Affiliations:** 1Zealand University Hospital, University of Copenhagen, 4200 Slagelse, Denmark; 2Department of Clinical Microbiology, Rigshospitalet, University of Copenhagen, 2100 Copenhagen, Denmark; 3Costerton Biofilm Center, Department of Immunology and Microbiology, University of Copenhagen, 2200 Copenhagen, Denmark

**Keywords:** biofilms, *Pseudomonas aeruginosa*, lung infection, immunology, inflammation

## Abstract

Biofilm infections are tolerant to the host responses and recalcitrance to antibiotic drugs and disinfectants. The induced host-specific innate and adaptive immune responses by established biofilms are significantly implicated and contributes to the course of the infections. Essentially, the host response may be the single one factor impacting the outcome most, especially in cases where the biofilm is caused by low virulent opportunistic bacterial species. Due to the chronicity of biofilm infections, activation of the adaptive immune response mechanisms is frequently experienced, and instead of clearing the infection, the adaptive response adds to the pathogenesis. To a high degree, this has been reported for chronic *Pseudomonas aeruginosa* lung infections, where both a pronounced antibody response and a skewed Th1/Th2 balance has been related to a poorer outcome. In addition, detection of an adaptive immune response can be used as a significant indicator of a chronic *P. aeruginosa* lung infection and is included in the clinical definitions as such. Those issues are presented in the present review, along with a characterization of the airway structure in relation to immune responses towards *P. aeruginosa* pulmonary infections.

## 1. Airways

Most invasive pathogens enter the body through mucosal surfaces. The airways can be regarded as the most exposed mucosal surface to microbes due to the daily inspiration of more than 10,000 L of non-sterile air, resulting in inhalation of microbes and the aspiration of microbes from the upper airways, including paranasal sinuses oral cavity and the pharynx [1,2]. Accordingly, normal air has been estimated to contain up to 10^5^ CFU/m^3^ of bacteria [3]. The present review on biofilm living *Pseudomonas aeruginosa* embraces two perspectives on the immune responses. Firstly, the anatomical angle discussing the conductive versus the respiratory airways and, secondly, the innate versus the adaptive immune response comprising time perspectives. The pulmonary airways can be divided into the conductive and the respiratory zone (Figure 1). In the conductive zone, the inspired air is heated and humidified and provide the passage for ventilation in the respiratory zone, where the O_2_ and CO_2_ exchange with the blood takes place in the alveoli. In addition, the conductive zone also harbors the most important mechanical innate defenses of the airways, including nasal hair, cilia and the mucus layer (the muco-ciliary escalator) with antimicrobial molecules such as defensins ensuring capturing and killing of the majority of inhaled microbes. This barrier function is effective and only a limited number of particles of small dimensions less then 2–5 µm can pass this barrier under normal circumstances [2]. The respiratory zone contains the alveolar macrophages, which are the most important pathogen-recognizing cells capable of phagocytosing microbes, which pervaded the barriers in the conductive zone, accordingly, maintaining homeostasis and initiating inflammatory responses across the narrow airway-blood barrier [1].

The conductive airways are a member of the mucosal associated lymphoid tissues (MALT), whereas the respiratory zones are associated to the systemic immune responses. The mucosal adaptive immune induction involves the nasal-associated lymphoid tissue (NALT) of the Waldeyers ring composed of the palatine, the lingual and the nasopharyngeal tonsils, including epithelium infiltrated with lymphocytes, and antigen-presenting cells and Microfold (M-)cells [4]. M-cells have also been identified in the lower respiratory tract in association to bronchus associated lymphoid tissue (BALT) in mice in a modified manner following inflammation or infection [5,6]. M-cells take up antigens from the luminal side and by transcytosis provide the antigens to immature dendritic cells (iDC). Similarly, tear duct-associated lymphoid tissue and conjunctiva associated lymphoid tissue is reported to be able to take up antigen and present a secretory IgA (sIgA) response (see later). Tear duct- and conjunctiva-associated lymphoid tissue can, together with the sublingual route, be important in vaccine development for mucosal immunity, but for established pulmonary biofilm infection, the significance is unknown. The overall MALT and the induction and ‘homing’ of effector functions have been extensively studied in gut-related responses, whereas the airway mucosal responses has so far been limited to IgA [4].

## 2. Biofilms

Biofilms are microbial communities embedded in a matrix composed of exopolymers. Biofilms formed by microorganisms that infect organs may cause recurrent and difficult to eradicate chronic infections, leading to a persistent inflammatory response which causes clinical challenges. Pulmonary biofilm infections depend on disposing patient factors in conditions such as cystic fibrosis (CF), or in patients with primary ciliary dyskinesia (PCD) or patients with diffuse panbrochiolitis, COPD or bronchiectasies [7]. Similarly, COVID-19 may predispose *P. aeruginosa* pulmonary biofilm infection as the SARS-CoV-2 infection creates a lung environment facilitating the adaptive characteristics of biofilm behavior [8,9]. Additional clinical situations being associated with respiratory biofilm infections are chronic sinusitis in CF or PCD patients, in allergies, in patients with anatomical abnormalities or immunodeficiencies or even in cases of gastroesophageal reflux [7]. Sinusitis can in some cases result in pulmonary infections due to aspiration to the lungs but will not be described further in the present review. The second additional clinical situation is observed in intubated patients where biofilms can form inside the endotracheal tubes and subsequently infiltrate the lungs (Figure 2). Accordingly, the pulmonary *P. aeruginosa* biofilm infections can either exist as chronic tissue-related biofilm infections in secretions of the airways or surface-related biofilms on the endotracheal tube, causing acute pulmonary infection upon bacterial release from the tubes into the airways [7]. The succeeding pulmonary damage is caused by a continuous inflammatory insult due to the activated host immune response. The distinction is highly important, since the disease pathogenesis, manifestations, clinical handling and immune responses are markedly different. For example, in acute pulmonary infections, virulence factors, such as type 3 secretion system (T3SS), which allow the translocation of bacterial effectors (ExoS, ExoU, ExoT and ExoY) directly into the cytoplasm of host cells, causing cytotoxicity or subversion of host defenses, play an important role. In contrast, biofilm infections select against T3SS-expression [10]. Although many CF patients carry antibodies against T3SS effector proteins [11], suggesting that these effector proteins were secreted at some stage of the infection, most *P. aeruginosa* strains isolated from chronic infection are T3SS negative. Loss of T3SS results in dampened inflammasome activation and reduced cell death in macrophages and neutrophils [12].

The consequences of biofilm formation encompass recalcitrance to antimicrobials and the effector functions of the immune system. Recently, the biofilm tolerance and resistance mechanisms were thoroughly reviewed and will not be described in detail in the present review [7].

## 3. Biofilm and Initiating the Immune System

The initial immune reaction to established biofilm infections is delivered by the innate immune system, which non-specifically attacks entering microorganisms [13,14]. During normal status, alveolar macrophages are approximately the only immune and inflammatory cell type present in the respiratory airways (Figure 2) [15,16,17]. In mice, alveolar macrophages are merely present in 40% of the alveoli, but compensate for the modest presence by being motile [1]. In situations of low exposure to bacteria the alveolar macrophages phagocytose and clear the pathogens (or aspirated particles) without initiating an infection. However, if the challenge dose surpasses 10^6^ CFU of planktonic living *P. aeruginosa*, polymorphonuclear leukocytes (PMNs) are increasingly being recruited from the blood system [1]. Whereas this threshold for *P. aeruginosa* is difficult to exceed under normal conditions with planktonic bacteria, aspiration of biofilm aggregated bacteria from the paranasal sinuses or infiltrated biofilms from tracheal tubes of ventilated patients may result in a focal high bacterial load [7,18]. Likewise, airway mucus stagnation as in CF can result in high bacterial concentrations in the lower airways. In these situations, the limited number of alveolar macrophages prevents removal of the pathogens and subsequent inflammation is initiated [1]. A simple, mechanistic process of protection is provided by the aggregation of several bacteria in biofilms, which increases the size of the object that PMNs must phagocytose, and biofilms with bacteria aggregates larger than 5 µm cannot be engulfed by the PMNs [19].

The lipopolysaccharide (LPS) of *P. aeruginosa* activates the complement system and attracts PMNs by factor C5a. Alveolar macrophages can recruit PMNs by pathogen recognition receptors (PRRs). PRRs bind to conserved pathogen-associated molecular patterns (PAMPs), which are microbe-associated molecular patterns (MAMPs) that can stimulate the innate immune response. For a long time, no biofilm-specific PAMPs were recognized. However, it has recently been suggested that molecular patterns lead to a stronger innate response when expressed in biofilms than when expressed in planktonic bacteria [20]. This subgroup of PAMPs has recently been identified in the biofilm matrix of *P. aeruginosa* and has been termed “biofilm-associated molecular patterns” (BAMPs) [20]. BAMPs may contribute to the immune-stimulatory properties of biofilms, and the specific exopolysaccharide composition of *P. aeruginosa* biofilms may determine the degree of respiratory burst and degranulation triggered by the responding PMNs [21]. Hence, it was discovered that BAMPs include the matrix exopolysaccharides alginate, Psl (polysaccharide synthesis locus) and Pel (pellicle), LPS [22] and filamentous Pf bacteriophages [20]. Recent in vitro studies have correspondingly demonstrated a direct interaction between *P. aeruginosa* biofilm matrix carbohydrates and different C-type lectins, suggesting that these receptors play a part in the immune response to *P. aeruginosa* infection [23].

Biofilm formation enables bacteria to proliferate, even though the innate immune response has been activated. Biofilm formation even provides protection when the host response is further reinforced by activation of the adaptive immune system, which involves the maturation and release of IgG and proinflammatory cytokines (infection stage 3, Figure 2) [7]. The mechanisms behind the immune tolerance are multifactorial and beyond the scope of this review, but was recently reviewed by O. Ciofu and colleagues [7].

The antibiotic therapy and immune response in chronic lung infection may result in a sufficiently reduced bacterial amount to escape immune recognition to survive, sometimes for many years, until the bacteria regrow and the clinical symptoms recur (infection stage 4, Figure 2).

## 4. Complement System

The complement system is likely to be activated during *P. aeruginosa* lung infections—both by means of the classical pathway and the alternative pathways. Complement system contribution has been determined by the frequent recognition of activated complement (C3c) in the sputum from CF patients with chronical pulmonary infections [24]. However, O-acetylation and polymerization of alginate has been linked to the protection of biofilm against the complement system [25,26].

## 5. T-Helper Cell Response

Innate lymphoid cells (ILC) have been reported to be important contributors to innate mucosal immunity and are divided into three subtypes according to their cytokine production; ILC1 (IFN-γ), ILC2 (IL-5, IL-9 and IL-13) and ILC3 (IL-17 and/or IL-22). Although they are lacking the T and B-cell receptors, ILC1 cells resemble Th1 cells and include NK cells, ILC2 are similar to Th2 cells and ILC3 resemble Th17 cells [4]. The inverse relation between ILC2 cells in peripheral blood of CF patients and disease severity and pulmonary failure indicates pulmonary homing of the IL2C cells [27]. ILC3 cells may enhance PMNs recruitment to the airways by means of IL-17A secretion [28], corresponding to adaptive T cell responses in CF (see below). The adaptive immune responses are effective in eliminating infections caused by non-sessile planktonic bacteria. In contrast, chronic biofilm infections result in persistence of the bacteria, since they cannot be eliminated completely. Instead, the synergy between the innate and adaptive immune responses, the latter with delay at first confront, is an essential biofilm pathogenesis element [21,29,30,31]. Dendritic cells (DC) are key initiators of the adaptive host responses and activation at the primary pathogen encounter [32]. Tissues harbor immature DCs, which are efficient in antigen uptake and are particularly copious at pathogen-exposed areas, such as mucosal surfaces and the secondary lymphoid tissue [33,34]. Succeeding antigen uptake and inflammatory cytokine control, DCs mature into committed and effective antigen-processing and antigen-presenting cells. The antigenic epitopes of biofilm that may induce T-helper cell immunity includes *P. aeruginosa* outer membrane protein (OMP) and LPS, as indicated by specific cytokine responses by peripheral blood mononuclear cells (PBMCs) from chronically infected CF patients [35,36].

Consequently, the crucial aspects of coupling the innate and adaptive immune systems are handled by DCs. In addition, DCs exclusively have the competence to prime naïve T-cells into succeeding Th1, Th2 or Th17 cells and subsequent responses [32,33,34]. Isolation and investigation of DCs is highly challenging due to the sparse occurrence of DCs in tissues, particularly for human studies. By means of a chronic *P. aeruginosa* lung infection model, we showed commitment of lung DCs during the infection as early as 2 days after infection onset [37]. Remarkably, the number of DCs in the regional lymph node did not increase until day 7. The fraction of activated pulmonary DCs increased during the 10-day observation period established by CD80 and CD86 expression, although the proportion of activated DCs in the lymph node diminished at day 10. DC cytokine production in the lung and lymph node were overall paralleled. Interestingly, however, the primary pro-inflammatory cytokines IL-6 and IL-12 reached a maximum at day 2–3, followed by an increased anti-inflammatory IL-10 production at day 7. We think this represents a significant controlling function of the DCs in initiation of the adaptive immune system effector functions, influenced by the adjacent innate reactions [37]. This observation is supported by reports of *P. aeruginosa* QS signal molecules reducing the murine DC IL-12 production but maintaining the IL-10 release. Furthermore, the antigen-specific T-cell proliferation was impaired by QS exposed DCs. The observations suggest that DCs are repressed during T-cell stimulation by the *P. aeruginosa* QS signals, and through this mechanism, add to the *P. aeruginosa* biofilm pathology [38,39].

The natural course of chronic *P. aeruginosa* lung infection displays a dichotomized outcome. Poor prognosis and a prominent or accelerated antibody response is observed for the majority of CF patients [40]. For a minor group of CF patients, however, the humoral response is modest and these patients have a favorable outcome. Furthermore, the intensified antibiotic treatment approach in CF, leading to considerably improved prognosis, actually associates to reduced antibody responses in CF [41]. A predominant Th2 immune reaction inducing a prominent humoral response was reported to correlate with the worse outcome of chronically infected CF patients [35,42]. In contrast, a predominant Th1 immune response was associated with better lung function in CF, presumably through a downregulation of the humoral response [35]. A modification from a Th2- to a Th1-dominated immune response was beneficial in animal studies, but a clinical study using inhalation IFN-γ failed in showing a positive effect in chronically infected CF patients [43]. To our knowledge, direct modulation of the adaptive immune response has not been performed against biofilm infections. By investigating specific cytokine release from re-stimulated PBMCs and later on cytokine measurements from unspecific stimulated T cells, a Th1/Th2 cytokine dichotomy in chronically infected CF patients was revealed [35,36]. Chronically infected CF patients had a Th2 dominated cytokine response with increased IL-4 (and IL-5, IL-10) production and diminished IFN-γ production. Moreover, a similar Th1/Th2 cytokine dichotomy was demonstrated in bronchoalveolar lavage fluid from CF patients [42,44]. Interestingly, IFN-γ release from PBMCs correlated to an improved lung function, suggesting a potential beneficial effect of IFN-γ [35]. Inbred mouse strains with chronic *P. aeruginosa* lung infection showed a pronounced pulmonary IFN-γ level in the relatively resistant C3H/HeN mouse strain [45,46]. Reinfection of the susceptible BALB/c mice resulted in a pulmonary Th1 response similar to the C3H/HeN mice and resembled the course of a primary infection in the C3H/HeN mice [47].

The explanation for the improved outcome of a Th1-dominated response in CF patients with chronic *P. aeruginosa* lung infection is probably that the Th2-dominated IgG response increases the inflammatory response and, thereby, the tissue damage [48]. A diminished Th2 response would presumably result in a reduced antibody response, due to reduced B- and plasma cell stimulation, and subsequently decreased immune complex formation and tissue damage. Important for the B-cell class switch is indeed the Th1 and Th2 cells, as stated. T-follicular helper (Tfh) cells contributes to this T-cell dependent B-cell response [49]. Tfh ‘home’ to the interface between T- and B-cell areas of the lymph node, where they can interact with recently activated B-cells and promote the differentiation into B memory cells and antibody-producing plasma cells (Figure 2). To our knowledge, the role of T fh has not been investigated in relation to *P. aeruginosa* pulmonary biofilm infections. Additional valid T cell subsets have been described, including the Th17 subset, characterized by the production of IL-17 and sometimes IL-22 [50]. Th17 cells are induced by TGF-β [51] and may be of interest in CF, since IL-17 induces the PMN mobilizer G-CSF and chemoattractant IL-8, thereby possibly adding to the pulmonary pathology of chronic *P. aeruginosa* lung infections [52,53]. In sputum from stable CF patients and in chronically infected CF, IL-17 and IL-23 was increased compared to CF patients without chronic *P. aeruginosa* lung infections [52]. Interestingly, such difference was not observed if the CF patients were infected with Staphylococcus aureus. In a later study, a substantially decreased fraction of peripheral Th17 cells in CF patients was reported and interpreted as augmented homing of the cells to the lungs, increasing the pulmonary inflammation [54]. In children with CF, the levels of both IL-17A and the Th2-related cytokines IL-5 and IL-13 were increased, but Th1-related cytokines were not, indicating a correlation between Th2 and Th17 subsets in CF [53]. Such a Th2-Th17 axis could dispose for *P. aeruginosa* lung infections, but this has not been clarified yet [42,53,55]. Interestingly, T cell-suppressive neutrophil myeloid-derived suppressor cells (MDSCs) have been reported in CF [56,57]. Presence of neutrophil MDSCs in peripheral blood correlated to improved lung function in CF and down-regulation of the harmful and dominating Th2 and Th17 response axis in CF could represent the underlying mechanism behind this observation [56].

sIgA vs. IgG in the airways IL-4 and TGF-β are important cytokines governing the isotype switch from surface (membrane bound) IgM-positive B cells to surface IgA-positive B cells and sIgA-producing plasma cells. Dendritic cells (DCs) can also induce surface IgA-positive B cells by stimulation with the B cell activation factor of the TNF family and a proliferation-inducing ligand. The differentiation into sIgA producing plasma cells is dependent on IL-5 and IL-6 [4]. sIgA1 is prominent in most mucosal tissues, whereas sIgA2 is mostly prevalent in the gut and genital tract. The transportation of sIgA and IgM is facilitated by binding to the epithelial polymeric immunoglobulin (Ig) receptor at the baso-lateral surfaces of epithelial cells, including the bronchial epithelial cells, nasal cavity, tonsils and tracheobronchial glands and by transcytosis transported into the mucosal secretions, whereafter sIgA is cleaved of the secretory component [4]. In the mucosal secretions, sIgA inhibits microbial bindings, neutralizes toxins and virulence factors and inhibits IgM and IgG mediated complement activation, and thereby, is anti-inflammatory and characterizes an ‘immune exclusion’. This also reduces levels of IL-6 and TNF-α and opsonization. The clinical significance of sIgA can especially be appreciated by the frequent airway infections in cases of IgA deficiencies. In the inherited disease cystic fibrosis (CF), sIgA has been found to be of significantly higher concentrations than other immunoglobulins and may also be produced earlier than IgG [58]. In CF patients, epithelial polymeric Ig receptors and IgA-producing plasma cells were increased in lung explants and levels of Pseudomonas-specific IgA were enhanced in sputum and serum [59]. In contrast, expression of the polymeric Ig receptor was decreased in COPD patients [60]. This may impact levels of sIgA in COPD patients, since stable patients non-colonized with *P. aeruginosa* had lower levels of sIgA than healthy controls, whereas COPD patients colonized by *P. aeruginosa,* still stable, increased their sIgA levels, but not above the levels in healthy controls [61]. In addition, COPD patients had subepithelial IgA accumulation and increased IgA expression in lung lymphoid follicles [62,63]. Alveolar cells of the respiratory zones can also produce sIgA; however, IgG responses dominates due to the transition from the blood stream in case of infection and subsequent inflammation. This results in complement activation and opsonization and increased inflammation and tissue damage.

The IgG responses are recognized as elements of the systemic immune response and primarily achieves access to mucosal surfaces through inflamed epithelium. sIgA is the primary antibody of mucosal surfaces and is produced in double the amount of IgG and secreted on mucosal surfaces as dimeric sIgA bound to the secretory component [64]. At the surfaces, sIgA functions through immune exclusion by binding to the pathogen and its PAMPs without activation of complement and opsonization [64]. In CF, sIgA is localized in sinuses and correlates to chronic sinusitis, whereas IgG dominates in the lower airways and corresponds to inflammation of the respiratory airways [65]. sIgA was also found to correlate to an early detection of *P. aeruginosa* of the lower airways of CF patients and a high negative predictive value to rule out chronic *P. aeruginosa* lung infection [65].

IgG, by means of precipitating antibodies in peripheral blood of CF patients, were associated with the appearance of the mucoid phenotype of *P. aeruginosa*. The antigens responsible for the specific IgG response include LPS, OMP and alginate—all of which are associated with the surface of *P. aeruginosa* biofilm [66,67,68,69]. A sequential development in specific antibody towards different *P. aeruginosa* antigens have also been reported. Indeed, a high or rapidly increasing humoral response correlates to a poorer prognosis [70,71,72]. The correlation between specific anti-pseudomonas IgG response and worse prognosis in CF probably has two key explanations. Firstly, the IgG antibody response has the capacity to activate the complement system and opsonize PMNs, and thus, due to inflammation accelerate the lung pathology in CF and probably also in other patients with chronic lung infections [18]. Secondly, both the affinity and avidity of the specific IgG response in CF decreases over time, which may still add to the collateral damage, but with reduced ability to bind firmly to the pathogens [18]. In contrast, the beneficial effect of the IgG response in CF have also been reported by the positive relationship of anti-β-lactamase IgG response and better lung function in CF [73]. This was supported by a vaccine study in a rat model of chronic *P. aeruginosa* lung infection, where the rats with the strongest anti-β-lactamase response also had the best outcome [73].

For a substantial number of CF patients, lung transplantation still remains a lifesaving option at end-stage lung disease. In a recent study, we evaluated the dynamics of the antibody response of 20 CF patients by quantifying precipitating antibodies one year before and up to five years after the transplantation [74]. All the patients experienced a significant drop in anti-*Pseudomonas* antibodies at one-year follow-up and no patients experienced the high antibody levels seen prior to lung transplantation. Interestingly, reinfection with *P. aeruginosa* had no effect on the antibody level. However, it remains to be clarified whether it is the removal of the substantial antigenic pulmonary load or the immune suppressive post transplantation treatments (or both) that leads to the substantial reduction in humoral response after lung transplantation in CF [74].

## 6. CFTR Modulators and Immune Response in CF

The novel class of treatment modalities in CF targeting the underlying protein defect has provided a giant step forward in managing the genetic disease [75]. Although debatable, we believe that the defect CFTR by itself do not result in inflammation, but the inflammatory stages in CF is induced by infection proceeding to chronic biofilm infection. This opinion is especially supported by the fact that CF patients do not suffer from increased infectious risk outside the airways. In a series of thorough, prospective investigations by Armstrong on newborn CF children [76,77,78], fetal CF lung tissue were histopathological normal and similar to non-CF fetal tissue [79]. The concept of infection preceding inflammation is supported by observations using the CF pig model [80]. Even though clinical and animal experimental data suggest no direct impact on inflammation, modulation and potentiation of the defect CFTR in the recruited PMNs (and macrophages) could add to the observed improved outcome in CF [81], since the membranes of the phagolysosomes contain CFTR [82,83]. Accordingly, avicaftor was able to restore degranulation of CF PMNs and improve their microbial killing [83]. Overall potentiator and modulator treatment evaluation is still premature and some results are conflicting. However, the observed treatment induced effect on mucocilliary clearance will presumably lead to reduced risk of recurrent and chronic lung infections and thereby reduced inflammation [81].

## 7. Prophylaxis of Pulmonary *Pseudomonas aeruginosa* Biofilm Infections

As evident from the challenges of treating biofilm infections and the failure of the immune system to eliminate established pulmonary biofilm infections, a prophylactic strategy is probably the most effective approach to combat pulmonary biofilm infections. Whereas antibiotic prophylaxis in CF is seemingly unsuccessful, the strategy of early antibiotic eradication therapy, pre-emptive treatment of colonization before it develops to chronic biofilm lung infections, is a universally used approach in cystic fibrosis patients intermittently colonized with *P. aeruginosa* [18]. This approach has been highly successful in CF and significantly delays and prevents the onset of chronic *P. aeruginosa* pulmonary biofilm infection in approximately 80% of the patients [75]. The approach is further supported by interventions diminishing the paranasal *P. aeruginosa* sinusitis of CF patients [84].

An intriguing approach for preventing *P. aeruginosa* pulmonary infection in CF patients was the passive immunotherapy using avian IgY immunoglobulins (yolk) targeting *P. aeruginosa*. IgY is the predominant serum antibody in chickens and is the avian homologue of mammalian IgG [85]. It accumulates in the egg yolk from the blood and provides the offspring with humoral immunity. Hyperimmunization of chickens with specific antigens provides high yields of specific IgY antibodies in the egg yolk [86]. The hypothesis was that CF patients gargling egg yolk from Pseudomonas-vaccinated hens would be protected by the anti-adhesive effect of the immunoglobulin, thereby potentially preventing the bacteria from colonizing the lower airways [87,88]. However, a thorough series of in vitro and in vivo studies revealed that both immune but also non-specific (näive) IgY rather functioned by aggregating *P. aeruginosa* enabling the PMNs to phagocytose the bacteria and preventing them from establishing biofilms in the lower airways [89,90,91]. Furthermore, the IgY effect could be augmented by adding azithromycin in a mouse model of *P. aeruginosa* lung infection [92]. A clinical strategy combining these two already implemented therapies in CF was tempting. However, an international multi-center RCT failed to show superior effect of anti-*Pseudomonas* IgY profylaxis compared to naïve IgY. Thus, anti-*P. aeruginosa* IgY for clinical use was arrested [93]. 

Active anti-Pseudomonas vaccination strategies have long been a motivated approach and current highly interesting possibilities emerges in novel vaccine development techniques. *P. aeruginosa* vaccines have been used successfully to protect farmed minks against acute *P. aeruginosa* pneumonia [94]. A comparison was made previously of the effect of a multi-component and a formalin-killed cell on protection against enzootic of hemorrhagic pneumonia due to *P. aeruginosa* in mink [95]. Vaccination against *P. aeruginosa* has also been investigated in CF patients using LPS, flagella or alginate as antigens [96,97,98]. Although a protective effect was observed concerning chronic biofilm infection, especially regarding the LPS and the flagella vaccines, the early antibiotic eradication therapy of intermittent *P. aeruginosa* infection was correspondingly efficient [99].

Another potential prophylactic treatment is bacteriophage therapy by administration of bacterial viruses that have lytic activity on *P. aeruginosa*. Due to their strain specificity, fast development of resistance to phages and to the recruitment of other bacterial defense mechanisms such as induction of alginate production, different combination of several phages (cocktails) with anti-pseudomonal activity are currently assessed in several phase I/II clinical trials in people with CF (clinicaltrials.gov). However, the role of the host immune response to phages is not clear, and at the present time, bacteriophage treatment is not available, in spite of promising in vitro results.

## 8. Conclusions

The crucial role of the immune response in the interface between infectious biofilm growing pulmonary *P. aeruginosa* and their hosts is increasingly being recognized for offering therapeutic pathways. The presence of specific precipitins against *P. aeruginosa* has long been appreciated for its diagnostic values and the correction of the deficient mucosal secretory response by CFTR modulators is a very promising new treatment modality that may change the influence of the immune response on the pulmonary infectious *P. aeruginosa* biofilm and the prognosis. This crossroad in CF will be followed closely by clinicians and researchers.

## Figures and Tables

**Figure 1 biomedicines-10-02064-f001:**
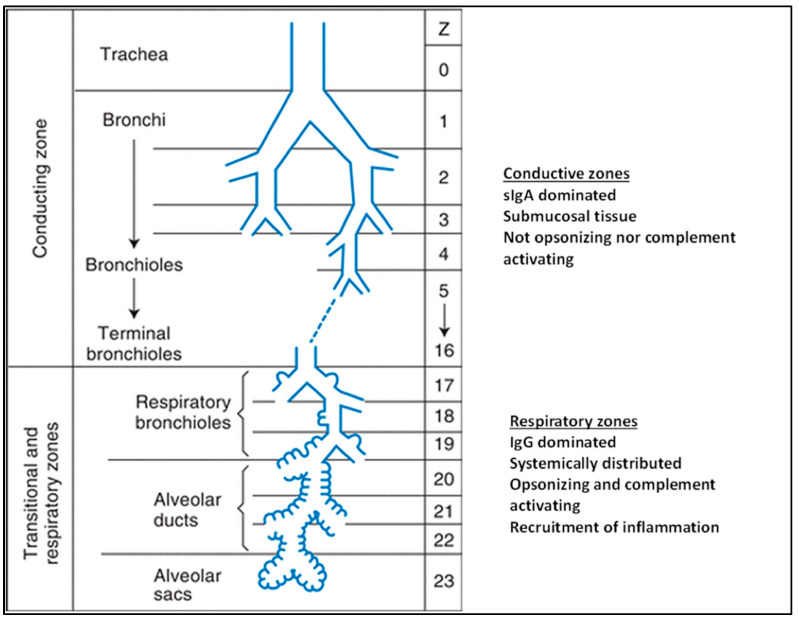
Illustration of the dichotomously branching of the airway from the conducting to the transitional and respiratory airway zones. The functions as well as the localized immune response of the conducting and respiratory zones differ. Numbers indicate approximate airway passages following generational branching (reproduced with permission from Ref. [2]. Copyright 2021 Wolters Kluwer.

**Figure 2 biomedicines-10-02064-f002:**
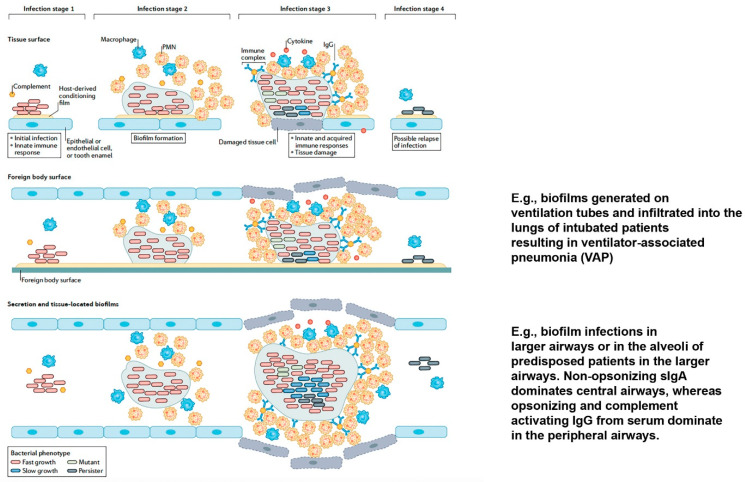
Graphic presentation of the tolerance of microbial biofilms to host immune responses according to the host-microbial biofilm interactions of biofilms on tissue surfaces (top panel), foreign body surfaces (middle panel) and within secretion and tissues (lower panel). The inflammatory response is demonstrated according to the distribution of macrophages and polymorphonuclear leukocytes (PMNs) in each of the four stages. Inflammation is also composed of the release of reactive oxygen species, proteases and DNA from PMNs (not indicated). In stage 1, resident macrophages and the complement system recruit PMNs and trigger an innate immune response following infection. During biofilm formation in stage 2, PMNs become the dominant cell. Stage 3 is characterized by tissue damage caused by an accelerated recruitment of proteolytic PMNs due to a synergic interplay between the innate immune response and the continual maturation of the adaptive immune response. The matured biofilm is characterized by a heterogenic bacterial population consisting of bacterial cells with variable growth rates, dormant and persister cells (slow growing), which are tolerant to the immune response. Mutants, including antibiotic-resistant isolates, also occur. The host response may effectively diminish the pathogenic biofilm and the subsequent dispersed bacteria to low levels, whereby the continual stimulation of the immune response and succeeding systemic signs of infection is annulled (stage 4) (reproduced with permission from Ref. [7], copyright 2022 Springer Nature).

## Data Availability

Not applicable.

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
