# Peer review of "Immune Response to Biofilm Growing Pulmonary *Pseudomonas aeruginosa* Infection"

_biomedicines, 2022, doi:10.3390/biomedicines10092064_

Round 1

Reviewer 1 Report

This review summarizes the host immune response to Pseudomonas aeruginosa biofilm grown during its infection in the lung. The authors discuss biofilm and how it initiates the immune system in the host. This is a significant part of this review. Overall this review is well structured and written. I recommend publishing. However, I have some minor suggestions.

1, For Figures 1 and 2, the graph should be drawn in a vector instead of a pixel-based graph.

2, Line 31, 105 instead of 105

3, Line 43, µm

4, Fig 2, at the left corner, bacterial phenotype, mutant means antibiotic resistant mutant. Also, the persister cell has a character of slow growth. 

5, Line 123, 106, the bacterial strain should be written in italic

6, Line 140 and 143, add the reference, please.

7, Line 172, interferon-gamma (IFN-γ)

8, Line 299, Reference.

9, Line 318, Mistyping.

10, Line 315, in the section of prophylaxis, it is surprising that the authors did not include bacteriophage therapy against P. aeruginosa biofilm during infection.

11, Although the T3SS is not required for the infection, the T3SS and relative effector proteins partially induce the host immune response. It would be better if the review covers T3SS and phage therapy.

Author Response

Dear Editor-in-Chief, Biomedicines.

Thank you for the fast review of our manuscript #... entitled Immune response to biofilm growing pulmonary Pseudomonas aeruginosa infection.

Following is a point-point response to the useful and relevant comments from the two reviewers:

Reviewer 1:

This review summarizes the host immune response to

Pseudomonas aeruginosa biofilm grown during its infection

in the lung. The authors discuss biofilm and how it initiates

the immune system in the host. This is a significant part of

this review. Overall this review is well structured and

written. I recommend publishing. However, I have some

minor suggestions.

Thank you

1, For Figures 1 and 2, the graph should be drawn in a

vector instead of a pixel-based graph.

The figures have been upgraded substantially in the manuscript. Since text is included in the figures from the source a vector format is not possible (for us) – we hope the present version is acceptable, or the your technical division can manage this issue.

2, Line 31, 10 instead of 105

This has been changed in the manuscript (including other places with a similar error.

3, Line 43, μm

Sorry, probably happened during the uploading process – has been changed

4, Fig 2, at the left corner, bacterial phenotype, mutant

means antibiotic resistant mutant. Also, the persister cell

has a character of slow growth.

We included these informations in the legends – we cannot change the figure itself, since this is from another publication. Please be aware, that other mutations than antibiotic resistance, also occurs – therefor the chosen formulation.

5, Line 123, 10 , the bacterial strain should be written

in italic

Has been changed, including several other places – may have happened during upload or other technical handling.

6, Line 140 and 143, add the reference, please.

We agree – ref 19 is also added here.

7, Line 172, interferon-gamma (IFN-γ)

Has been changed

8, Line 299, Reference.

Ref 61 also added here.

9, Line 318, Mistyping.

b was inserted, including the line to lines beneath

10, Line 315, in the section of prophylaxis, it is surprising

that the authors did not include bacteriophage therapy

against P. aeruginosa biofilm during infection.

We agree and have included a section on phages, although at the end before the concluding section.

11, Although the T3SS is not required for the infection, the

T3SS and relative effector proteins partially induce the host

immune response. It would be better if the review covers

T3SS and phage therapy.

We also included a section of T3SS in the ‘Biofilm’ paragraph.

Reviewer 2:

The article is written in a professional and skilled language.

There is a treasury of knowledge on a given topic and can

provide substantial substantive support for students and

have taught academic teachers in this field. The work

responds to all the assumptions and is perfectly described

in the result part as well as in the discussion, therefore it

requests its approval in its present form, as it is.

Thank you.

On behalf of all co-authors

Sincerely yours

Claus Moser

Reviewer 2 Report

I accept in the present form

The article is written in a professional and skilled language. There is a treasury of knowledge on a given topic and can provide substantial substantive support for students and have taught academic teachers in this field. The work responds to all the assumptions and is perfectly described in the result part as well as in the discussion, therefore it requests its approval in its present form, as it is.

Author Response

(The authors gave the same response as above.)
